

# Influence of 4-week lower extremity high-intensity interval training on energy metabolism and maximal oxygen uptake of elite swimmers

Rangxi Jin[1], Junhui Sun[1], Na Jiang[2] and Chao Chen[3]

[1] School of Athletic Performance, Shanghai University of Sport, Shanghai, Shanghai, China
[2] Medicine and Sports Health Promotion Medical College, Dalian University, Dalian, Liaoning, China
[3] Dalian University College of Physical Education, Dalian University, Dalian, Liaoning, China

Corresponding author
Chao Chen, chenchao@dlu.edu.cn

## ABSTRACT

**Objectives.** This study examines the effects of a 4-week high-intensity interval training (HIIT) program on energy metabolism and maximal oxygen uptake ($VO_2$max) in elite swimmers, aiming to provide empirical evidence for optimizing competitive swimming training.

**Methods.** Twenty-four competitive swimmers were randomly assigned to either an HIIT experimental group or a control group. The experimental group underwent a structured 4-week lower-limb HIIT program, while the control group continued their regular training regimen. Energy metabolism parameters and $VO_2$max were assessed using a Lode lower-limb power cycle and a gas metabolism analyzer. Repeated measures analysis was used to examine interaction effects, with data analysis conducted at a significance level of $P < 0.05$.

**Results.** The HIIT group exhibited significant improvements in all energy metabolism parameters and $VO_2$max. Phosphagen energy supply increased from $40.39 \pm 9.46$ kJ to $58.27 \pm 9.12$ kJ ($P < 0.001$), glycolytic energy supply increased from $41.81 \pm 9.81$ kJ to $59.06 \pm 10.86$ kJ ($P < 0.001$), and aerobic energy supply increased from $132.29 \pm 25.12$ kJ to $173.32 \pm 29.50$ kJ ($P < 0.001$). Consequently, total energy supply rose from $214.48 \pm 38.58$ kJ to $290.65 \pm 42.01$ kJ ($P < 0.001$). Additionally, $VO_2$max significantly improved from $51.48 \pm 3.85$ ml/min/kg to $55.03 \pm 4.90$ ml/min/kg ($P = 0.041$), whereas no significant changes were observed in the control group.

**Conclusion.** The findings confirm that a 4-week lower-limb HIIT program significantly enhances energy metabolism and $VO_2$max in elite swimmers. These results underscore the efficacy of HIIT in improving metabolic adaptability, thereby supporting its application as a key training strategy for optimizing competitive swimming performance.

# INTRODUCTION

Competitive swimming performance results from the complex interplay of multiple factors, including energy metabolism, technical execution, psychological resilience, and

physiological conditioning (*Dalamitros et al., 2016*; *Sousa et al., 2018*; *Zacca et al., 2020*; *Zamparo, Cortesi & Gatta, 2020*; *Sammoud et al., 2021*; *Song & Sheykhlouvand, 2024*). Among various training methods aimed at enhancing these components, high-intensity interval training (HIIT) has gained considerable attention for its efficiency and physiological benefits. HIIT alternates periods of near-maximal exertion (typically 80–100% $VO_2$max or HRmax) with intervals of rest or low-intensity activity, thereby stimulating both aerobic and anaerobic systems (*Sperlich et al., 2010*; *Gillen & Gibala, 2014*; *Moniz, Islam & Hazell, 2020*). Numerous studies have demonstrated that HIIT is effective in improving swimming performance, particularly through enhancing aerobic capacity, anaerobic power, and energy efficiency (*Sperlich et al., 2010*; *Dalamitros et al., 2016*; *Clemente-Suárez & Arroyo-Toledo, 2018*; *Weng & Hou, 2020*; *Massini et al., 2023*). Sprint interval training (SIT) has also gained popularity due to its similar effectiveness and reduced time demands (*Eigendorf et al., 2021*). For swimmers, HIIT is a preferred training modality not only for aerobic enhancement but also for its ability to complement technical training such as stroke mechanics, starts, and turns.

From a bioenergetic perspective, HIIT imposes demands on all three primary energy systems: the phosphagen system, anaerobic glycolysis, and aerobic metabolism (*Gastin, 2001*). Training protocols such as 90/30 and 30/30 manipulate the work-to-rest ratio to modulate these energy pathways, influencing both lactate accumulation and oxygen uptake (*Gosselin et al., 2012*; *Almeida et al., 2021*). Compared with moderate-intensity training, HIIT more effectively improves repeated sprint ability (RSA), enhances metabolic recovery, and reduces fatigue-inducing byproducts (*Edge et al., 2005*; *Norberto et al., 2021*). Such adaptations are especially important for events ranging from 50 m to 400 m, where energy system interplay is critical (*Gonjo et al., 2018*). Furthermore, tailoring HIIT programs to emphasize aerobic, anaerobic, or mixed metabolic responses offers coaches a powerful tool for targeted performance development (*Ponimasov & Bolotin, 2019*).

Despite consistent findings supporting HIIT's physiological benefits, results regarding its influence on $VO_2$max in elite athletes remain mixed. Some studies suggest that when total training volume is significantly reduced, as in *Kilen et al. (2014)*, HIIT may not elicit significant $VO_2$max gains, possibly due to the athletes' already optimized cardiovascular capacity. Others, however, report $VO_2$max improvements of 3–4% even within short durations (*Caputo & Denadai, 2008*; *Menz et al., 2015*; *Weng & Hou, 2020*). Training mode also plays a role; for example, Twist and Wist (*Twist, Bott & Highton, 2023*) found that running-based HIIT induced greater cardiovascular load than cycling-based HIIT. Moreover, these improvements appear to be mediated by peripheral adaptations, such as increased stroke volume, rather than changes in hemoglobin mass or blood volume (*Menz et al., 2015*). The effectiveness of HIIT and SIT depends on factors such as protocol design, training status, and individual variability (*Sloth et al., 2013*; *Del Giudice et al., 2020*; *Rosenblat, Granata & Thomas, 2022*).

However, a significant gap remains in the literature regarding HIIT's localized effects on swimmers' lower-limb energy metabolism and neuromuscular adaptation. While upper-body propulsion dominates in swimming, lower-limb contributions to starts, turns, and kicking patterns are critical for competitive success. Most existing studies have focused on

systemic responses, overlooking the specific adaptive changes in lower-limb musculature. Thus, this study aims to investigate the effects of a 4-week HIIT intervention targeting the lower limbs of elite swimmers. Specifically, we examine how variations in training intensity and interval structure affect the activation of phosphagen, glycolytic, and aerobic energy systems, as well as VO$_2$max. We hypothesize that a 4-week lower-limb HIIT intervention will significantly enhance the energy metabolism—specifically phosphagen, glycolytic, and aerobic energy systems—as well as maximal oxygen uptake (VO$_2$max) in elite swimmers, compared to a control group maintaining regular training.

## MATERIALS AND METHODS

### Experimental approach to the problem

The testing and intervention at the Physical Training Research Center of Shanghai University of Sport. All participants completed the lower-limb strength cycling tests within the same week, with at least 24 h between the two tests (*Funai et al., 2025*). The ideal follow-up testing time for each participant was the same as the initial test time. Before all tests were conducted, height and weight measurements were taken to assess the athletes' height, weight, and BMI, ensuring the accuracy of the VO$_2$max values. This study was approved by the Ethics Committee of Shanghai University of Sport (102772021RT031), and all participants signed informed consent forms. The research roadmap is as follows Fig. 1.

### Participants

*A priori* sample size estimation was conducted using G*Power 3.1, with parameters set at a medium effect size ($F = 0.25$), significance level $\alpha = 0.05$, and statistical power ($1-\beta$) = 0.80. The analysis indicated that a minimum of 24 participants was required. Accordingly, 30 elite swimmers from Shanghai University of Sport were initially recruited. After screening, six individuals were excluded—three for not meeting the inclusion criteria, two for declining to participate, and one due to injury risk. A total of 24 athletes were formally enrolled in the study and randomly assigned in a 1:1 ratio to either the experimental group (undergoing lower-limb high-intensity interval training) or the control group (maintaining regular training), using a computer-generated random sequence by an independent researcher. Each group consisted of 12 participants, including eight males and four females. All participants successfully completed the 4-week intervention and post-test assessments.

Eligibility criteria required participants to have a minimum of 10 years of competitive swimming experience and no history of injury or illness within the previous six months. Exclusion criteria included chronic diseases, recent surgeries, cardiovascular risks, or current use of performance-enhancing substances. All participants' health status was confirmed *via* self-reported medical history and screening by a certified sports physician.

The physical characteristics of the participants were as follows (mean $\pm$ SD): lower-limb HIIT group—age, 19.94 $\pm$ 1.4 years; height, 176.08 $\pm$ 5.4 cm; weight, 68.32 $\pm$ 6.3 kg; training experience, 13.00 $\pm$ 1.5 years; control group—age, 21.17 $\pm$ 1.6 years; height, 177.54 $\pm$ 5.8 cm; weight, 73.95 $\pm$ 6.9 kg; training experience, 11.67 $\pm$ 1.7 years.

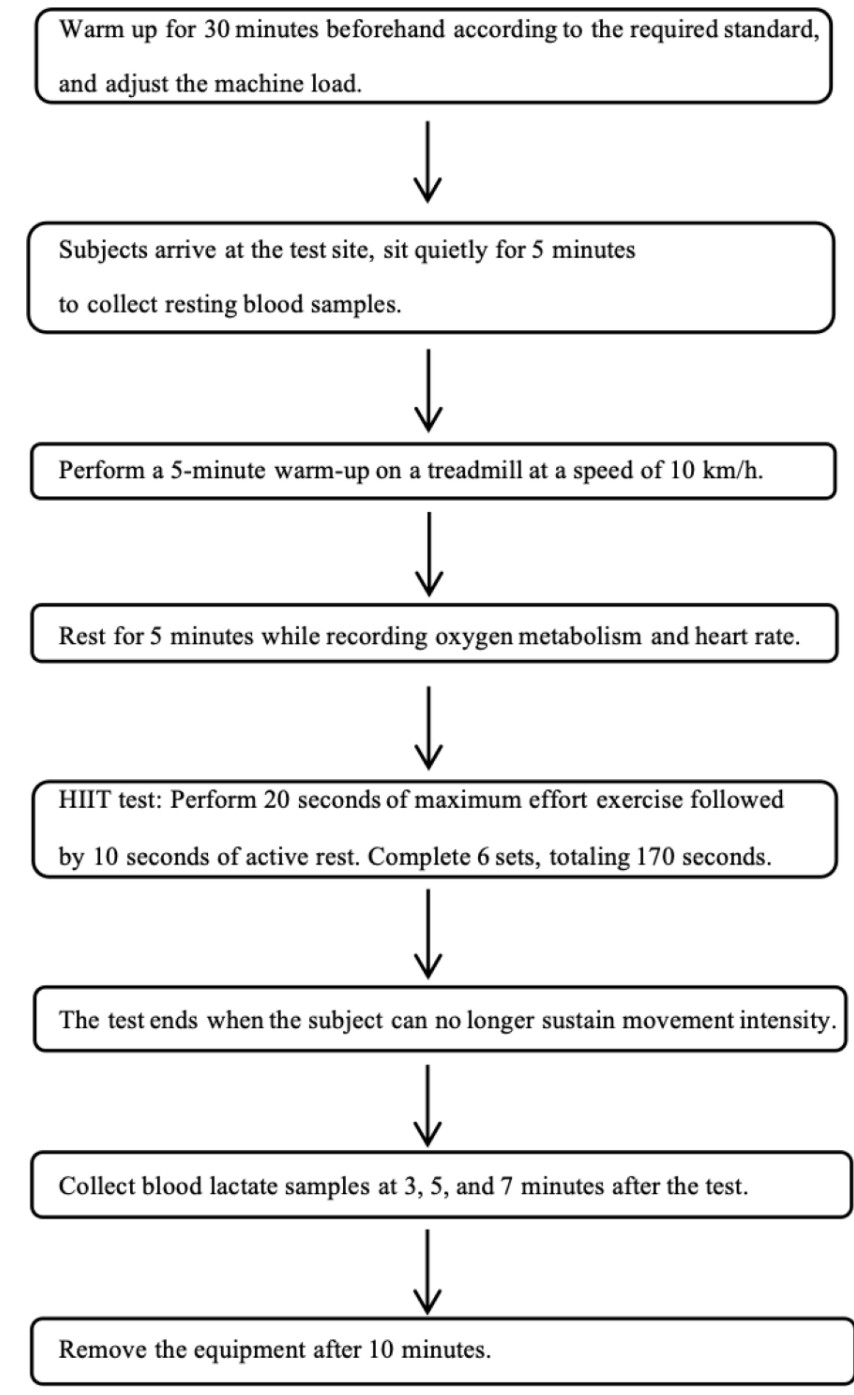

**Figure 1  Lower-limb HIIT testing flowchart.**

Each group included four female participants. The menstrual cycle phase of female athletes was not monitored or controlled during the intervention. Although previous studies have shown that performance may fluctuate between the follicular and luteal

phases, this variable was not accounted for in the current study and should be addressed in future investigations (*Archacki et al., 2024*).

## Experimental intervention plan and procedure

The 4-week high-intensity interval training (HIIT) program is summarized in Table 1. The high-intensity interval training (HIIT) intervention consisted of eight dry-land exercises targeting lower-limb muscular power and coordination: jump squats, alternating lunges, bounding, single-leg hops, lateral jumps, split squat jumps, wall sits with isometric pulses, and quick-feet drills. All exercises were performed on land in a standardized circuit sequence to ensure consistency of stimulus. Each training session comprised three full circuits, with each exercise performed for 20 s at maximal intensity followed by 10 s of rest, totaling 230 s per circuit. Two sets were completed per session, separated by a 3-minute rest interval. The entire session lasted approximately 60 min, including a 20-minute standardized warm-up, 30 min of HIIT work, and 10 min of post-training stretching and relaxation.

The program was carried out over four weeks, with three sessions per week, for a total of 12 training sessions. The 4-week duration was selected based on prior HIIT research demonstrating significant metabolic and cardiorespiratory adaptations following short-term mesocycles. This timeline is also practical for integration into elite swimmers' in-season training without disrupting technical and in-water workloads.

The selected exercises were informed by a literature-based approach and chosen for their specificity to lower-body movement patterns relevant to swimming performance. Prior studies have emphasized the importance of strength and power development through dry-land resistance training in swimmers (*Bishop et al., 2013*; *Crowley, Harrison & Lyons, 2018*). In addition, plyometric and dynamic stability movements, such as those used in this protocol, have been shown to improve sport-specific actions including push-off, kicking propulsion, and start/turn explosiveness (*Sammoud et al., 2021*).

The lower-limb cycling power test was selected to assess energy metabolism and $VO_2$max due to its practicality and proven validity in evaluating lower-body aerobic and anaerobic performance. While swimming-specific tests conducted in water can provide direct sport-specific insights, they are limited by technical challenges, such as respiratory interference, lack of standardized protocols, and equipment constraints for measuring metabolic variables. In contrast, cycling-based protocols enable precise control over workload, reliable data acquisition, and high repeatability, making them suitable for evaluating lower-limb energy system responses in elite swimmers.

## HIIT energy metabolism test
### Test instrument

The following equipment were used in this study: a Lode lower-limb cycle ergometer (Lode Excalibur Sport; Lode BV, Groningen, Netherlands; Fig. 2) for aerobic/anaerobic power testing, a K5 portable metabolic system (Cosmed K5, Cosmed, Rome, Italy) for gas exchange analysis, a Polar Accurex Plus heart rate monitor (Polar Electro Oy, Kempele, Finland), and an EKF BIOSEN S-line lactate analyzer (EKF Diagnostic, Barleben, Germany). Additional materials included a Borg RPE scale, sterile blood collection needles, alcohol swabs, EKF sample tubes, medical gloves, a stopwatch, and markers.

**Table 1  High-intensity interval training program.**

| Plan | Content | Load | Class number | Exercise time/s | Interval time/s |
|---|---|---|---|---|---|
| | High leg lift | Be self-possessed | 3 | 20 | 10 |
| | Prone elastic band open and close | – | 3 | 20 | 10 |
| | BOSU squat and jump from side to side | – | 3 | 20 | 10 |
| Lower extremity | Standing jumping with elastic band | Blue elastic ring | 3 | 20 | 10 |
| | Retraction leg squat jump | Be self-possessed | 3 | 20 | 10 |
| | Prone slide board | Be self-possessed | 3 | 20 | 10 |
| | Left- and righ- side jump stool | Be self-possessed | 3 | 20 | 10 |
| | Elastic band resists longitudinal jump | Be self-possessed | 3 | 20 | – |

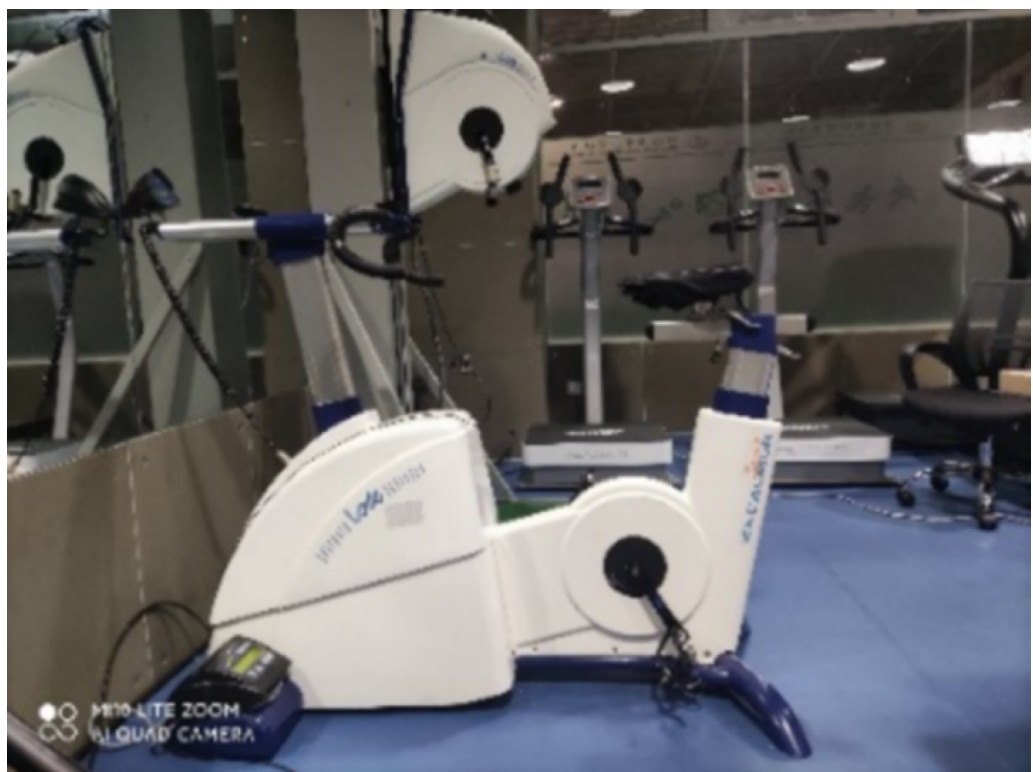

**Figure 2  Photograph of lower-limb power bicycle.** Image credit: Strength and Conditioning Center of Shanghai University of Sport.

### Test mode

Before the lower-limb power cycle test was conducted, the seat and handlebar positions of the power cycle were adjusted on the basis of the participant's subjective comfort, ensuring that the knee joint remained slightly bent when the leg was fully extended. The training load on the Lode lower-limb power cycle was set at 4.5% of the participant's body weight. For example: T = body weight × 4.5% × 9.8 × torque = @N.M. This study modeled the HIIT test after the Tabata protocol, with continuous testing and verbal encouragement provided throughout the process to maximize the athlete's enthusiasm.
*Procedure*

The K5 device was preheated for 30 min and calibrated in accordance with the manufacturer's requirements, including pressure, gas composition (standard gas concentrations: 15.00% $O_2$, 5.00% $CO_2$, and 80.0% $N_2$), and gas volume (3 L gas tank). Each participant selected an appropriate mask size prior to the first test, and the same type of mask was used for all four tests. Gas collection, storage, and analysis were conducted using standard software (Mate Soft Cosmed Italy). The treadmill was calibrated prior to the test. Upon arrival at the testing site, the participants rested for 5 min while their resting blood lactate levels were collected (20 µL from the earlobe) and the experiment procedures, precautions, and RPE scale were explained. A warmup of 5 min at 10 km/h was performed on the treadmill, followed by 5 min rest, during which the calibrated gas analyzer was fitted. Before the test, the participants were instructed on how to use the RPE scale, which includes 15 levels representing fatigue from mild to severe. The RPE scores were recorded before and after the test. The test began with participants performing 20 s of all-out pedaling, followed by 10 s rest. Each participant completed six sets within 170 s. Blood lactate samples (20 µL from the earlobe) were collected and analyzed using a lactate analyzer at 3, 5, 7, and 10 min before and after the test. After the test, the K5 device was removed 10 min post-exercise, during which the participants sat on a bench for passive recovery and were reminded to minimize speaking.

*Data operation*

The aerobic, glycolytic, and phosphagen energy supplies during all-out exercise were calculated using accumulated oxygen uptake, accumulated blood lactate, and the rapid phase of oxygen debt post-exercise. Phosphagen energy supply was determined as the rapid phase of oxygen debt (mL) multiplied by the energy equivalent (J/mL). Glycolytic energy supply was calculated using accumulated blood lactate (mmol), the oxygen-lactate conversion factor (3.0 mL/mmol/kg), body weight (kg), and the energy equivalent (J/mL). Aerobic energy supply was derived from accumulated oxygen uptake (mL) multiplied by the energy equivalent (J/mL). The rapid phase of oxygen debt was computed as the actual oxygen uptake during the first 3 min post-exercise minus the inverted oxygen uptake curve (slow phase) during the subsequent 3 min: Rapid phase of oxygen debt = actual oxygen uptake (first 3 min)—slow phase (first 3 min). When the respiratory exchange ratio (RER) exceeded 1.0, the energy equivalent was set at 21.131 J/mL of oxygen consumed (*Stegmann, 1977*). In this study, resting oxygen uptake values were standardized as 3.5 mL/min/kg for females and 4.0 mL/min/kg for males.

## Statistical analysis

All statistical analyses were conducted using SPSS (Version 25.0; IBM Corp., Armonk, NY, USA), while Microsoft Excel (Version 2010) was used solely for data management and visualization. Descriptive data are presented as mean ± standard deviation (M ± SD).

To evaluate the effects of the intervention, repeated-measures ANOVA was employed to assess within-subject (time: pre *vs.* post), between-group (HIIT *vs.* control), and interaction (group × time) effects. One-way ANOVA was used for between-group comparisons, and paired-sample t-tests were applied for within-group pre-post comparisons.

In addition to $p$-values, effect sizes were calculated using partial eta squared ($\eta^2$) for all main and interaction effects. Interpretation of $\eta^2$ followed Cohen's conventions: trivial (<0.2), small (0.2–0.5), moderate (0.5–0.8), and large (>0.8), to better reflect the practical significance of the results. Statistical significance was set at $P < 0.05$.

## RESULTS

### Results of lower limb HIIT intervention on energy metabolism

The lower-limb training group demonstrated significant improvements in energy metabolism supply across four dimensions: phosphagen, glycolytic, aerobic, and total energy supply (Table 2). After training, the phosphagen energy supply increased significantly from $40.39 \pm 9.46$ kJ to $58.27 \pm 9.12$ kJ ($F = 21.930$, $P < 0.001$, $\eta^2 = 0.499$), the glycolytic energy supply increased from $41.81 \pm 9.81$ kJ to $59.06 \pm 10.86$ kJ ($F = 45.964$, $P < 0.001$, $\eta^2 = 0.676$), the aerobic energy supply increased from $132.29 \pm 25.12$ kJ to $173.32 \pm 29.50$ kJ ($F = 36.538$, $P < 0.001$, $\eta^2 = 0.624$), and the total energy supply increased from $214.48 \pm 38.58$ kJ to $290.65 \pm 42.01$ kJ ($F = 72.413$, $P < 0.001$, $\eta^2 = 0.767$). By contrast, the control group showed no significant changes in any energy supply dimension ($P > 0.05$). Significant effects were observed for time, group, and their interaction, with interaction effects particularly pronounced for total energy supply ($F = 101.979$, $P < 0.001$, $\eta^2 = 0.823$), indicating a strong regulatory effect of training on the lower-limb energy metabolism system. However, the relative contribution percentages of the energy supply pathways (phosphagen %, glycolytic %, anaerobic %, and aerobic %) did not show significant changes post-training ($P > 0.05$). This finding suggests that the training primarily enhanced the total energy supply rather than altering the relative distribution of energy metabolism pathways.

The lower-limb group exhibited significant differences in group interaction effects, particularly in glycolytic energy supply ($F = 105.836$, $P < 0.001$, $\eta^2 = 0.828$) and total energy supply ($F = 173.130$, $P < 0.001$, $\eta^2 = 0.887$) (Table 3). These results highlight that training substantially enhanced the energy supply capacity of metabolic pathways. By contrast, the control group showed significant effects in time interaction (*e.g.*, glycolysis: $F = 28.486$, $P < 0.001$, $\eta^2 = 0.564$; total energy supply: $F = 46.174$, $P < 0.001$, $\eta^2 = 0.677$) but did not demonstrate significant differences in group interaction effects ($P > 0.05$). This finding indicates that without intervention, the energy supply did not undergo substantial changes. Overall, the significance of time and group interaction effects further confirmed the substantial impact of training on the lower-limb group's energy supply across different metabolic pathways. The effects were particularly pronounced in glycolysis and total energy supplies, providing scientific evidence for optimizing exercise training programs.

### Results of lower-limb HIIT intervention on VO$_2$max

Our analysis of maximal oxygen uptake (VO$_2$max) revealed significant training effects (Table 4). Following the intervention, the lower-limb group demonstrated a clinically meaningful 6.9% improvement in VO$_2$max (pre: $51.48 \pm 3.85$ ml/min/kg *vs.* post: $55.03 \pm 4.90$ ml/min/kg; $P < 0.05$), while the control group showed no significant changes (pre: $49.40 \pm 4.12$ ml/min/kg *vs.* post: $48.19 \pm 3.94$ ml/min/kg; $P > 0.05$).

Jin et al. (2025), *PeerJ*, DOI 10.7717/peerj.19788

**Table 2  Energy supply and statistical analysis of different energy metabolic pathways in the lower-limb and control groups before and after training.**

| Energy metabolic | Lower extremity group | | Control group | | F-value | | | P-value | | | η2 | | |
|---|---|---|---|---|---|---|---|---|---|---|---|---|---|
| | Pre | Post | Pre | Post | Time | Group | G × T | Time | Group | G × T | Time | Group | G × T |
| Phosphoric acid/kJ | 40.39 ± 9.46 | 58.27 ± 9.12[*,#] | 39.29 ± 10.32 | 36.73 ± 7.91 | 21.930 | 11.089 | 38.979 | 21.930 | 11.089 | 38.979 | 0.499 | 0.335 | 0.639 |
| Glycolysis/kJ | 41.81 ± 9.81 | 59.06 ± 10.86[*,#] | 38.47 ± 10.31 | 37.29 ± 9.03 | 45.964 | 60.362 | 60.362 | 45.964 | 60.362 | 60.362 | 0.676 | 0.318 | 0.733 |
| Aerobic/kJ | 132.29 ± 25.12 | 173.32 ± 29.50[*,#] | 121.62 ± 21.89 | 118.84 ± 17.82 | 36.538 | 12.376 | 47.912 | 36.538 | 12.376 | 47.912 | 0.624 | 0.360 | 0.685 |
| Total energy supply/kJ | 214.48 ± 38.58 | 290.65 ± 42.01[*,#] | 199.37 ± 34.47 | 192.87 ± 26.83 | 72.413 | 16.067 | 101.979 | 72.413 | 16.067 | 101.979 | 0.767 | 0.422 | 0.823 |
| Phosphoric acid/% | 18.87 ± 3.45 | 20.10 ± 1.77 | 19.55 ± 3.24 | 19.15 ± 3.63 | 0.312 | 0.017 | 1.189 | 0.312 | 0.017 | 1.189 | 0.014 | 0.001 | 0.051 |
| Glycolysis/% | 19.41 ± 2.38 | 20.40 ± 3.10 | 19.21 ± 4.00 | 19.22 ± 3.56 | 1.821 | 0.279 | 1.763 | 1.821 | 0.279 | 1.763 | 0.076 | 0.013 | 0.074 |
| Anaerobic % | 38.28 ± 4.04 | 40.50 ± 3.06 | 38.76 ± 5.87 | 38.38 ± 3.57 | 1.762 | 0.263 | 3.558 | 1.762 | 0.263 | 3.558 | 0.074 | 0.012 | 0.139 |
| Aerobic/% | 61.72 ± 4.04 | 59.50 ± 3.06 | 61.24 ± 5.87 | 61.62 ± 3.57 | 0.553 | 0.444 | 1.116 | 0.461 | 0,509 | 0.297 | 0.012 | 0.009 | 0.024 |

**Notes.**

Values are presented as mean ± standard deviation (±SD).

Statistical analysis were performed using repeated-measures ANOVA for time, group, and group × time interactions.

$\eta^2$ represents partial eta squared, used to estimate effect size. Effect size interpretation follows Cohen's thresholds:

● Trivial (<0.2)   ● Small (0.2–0.5)   ● Moderate (0.5–0.8)   ● Large (>0.8).

[*]$P < 0.05$ *vs.* pre-test.

[#]$P < 0.05$ *vs.* control group.

**Table 3  Simple effect analysis of energy supply in different metabolic pathways for the lower-limb and control groups before and after training.** This table presents the simple effect analysis of energy supply across metabolic pathways, comparing pre- and post-training differences within and between groups. Data were analyzed using one-way ANOVA.

| Energy metabolism | Group | Time interaction | | | Group interaction | | |
|---|---|---|---|---|---|---|---|
| | | $F$ | $P$ | $\eta^2$ | $F$ | $P$ | $\eta^2$ |
| Phosphoric acid/kJ | Lower extremity | 0.074 | 0.788 | 0.003 | 59.691 | <0.001 | 0.731 |
| | Control | 38.240 | <0.001 | 0.635 | 1.217 | 0.282 | 0.052 |
| Glycolysis/kJ | Lower extremity | 0.660 | 0.425 | 0.029 | 105.836 | <0.001 | 0.828 |
| | Control | 28.486 | <0.001 | 0.564 | 0.490 | 0.491 | 0.022 |
| Aerobic/kJ | Lower extremity | 1.231 | 0.279 | 0.053 | 84.065 | <0.001 | 0.793 |
| | Control | 29.983 | <0.001 | 0.577 | 0.385 | 0.541 | 0.017 |
| Total energy supply/kJ | Lower extremity | 1.024 | 0.323 | 0.044 | 173.130 | <0.001 | 0.887 |
| | Control | 46.174 | <0.001 | 0.677 | 1.262 | 0.273 | 0.054 |

Notes.

$\eta^2$ values represent partial eta squared, a measure of effect size. Based on Cohen's guidelines:
- Trivial (<0.2)  • Small (0.2–0.5)  • Moderate (0.5–0.8)  • Large (>0.8).

$F$, test statistic; $P$, significance value; $\eta^2$, effect size.

A two-way repeated measures ANOVA indicated significant main effects for time ($F(1,22) = 4.738$, $P = 0.041$, $\eta^2 = 0.177$) and group ($F(1,22) = 8.174$, $P = 0.009$, $\eta^2 = 0.271$), with a particularly strong time × group interaction ($F(1,22) = 19.544$, $P < 0.001$, $\eta^2 = 0.470$). These results confirm that the lower-limb HIIT protocol specifically enhanced aerobic capacity, as quantified by $VO_2$max improvements in the experimental group that were absent in controls.

In terms of time interaction effects, the between-group difference before training (pre-test) was not statistically significant ($F = 1.808$, $P = 0.192$, $\eta^2 = 0.076$) (Table 5). However, the between-group difference after training (post-test) was significant ($F = 15.371$, $P < 0.001$, $\eta^2 = 0.411$), indicating a substantial difference in $VO_2$max between the groups following the intervention. Regarding group interaction effects, the lower-limb group exhibited significant differences before and after training ($F = 21.764$, $P < 0.001$, $\eta^2 = 0.497$), whereas the control group showed no significant changes ($F = 2.518$, $P = 0.127$, $\eta^2 = 0.103$). Overall, the lower-limb group demonstrated a significant improvement in $VO_2$max after training, whereas no notable changes were observed in the control group. These findings further confirm the effectiveness of lower-limb training in enhancing $VO_2$max and highlight the critical role of group and time interactions in training outcomes.

## DISCUSSION

This study systematically analyzed the responses of elite swimmers to 4 weeks of lower-limb HIIT intervention, focusing on changes across different energy metabolism pathways and $VO_2$max. The results showed that lower-limb HIIT significantly enhanced phosphagen, glycolytic, and aerobic energy supplies while improving $VO_2$max. Compared with the control group, the experimental group demonstrated significant advantages in energy metabolism and aerobic adaptability, confirming the effectiveness of lower-limb HIIT in

Jin et al. (2025), *PeerJ*, DOI 10.7717/peerj.19788

**Table 4** Maximal oxygen uptake (VO$_2$max) supply and statistical analysis results for the lower-limb and control groups before and after training.

| VO$_2$max/ (ml/min/kg) | Lower extremity | | Control group | | F | | | P | | | η2 | | |
|---|---|---|---|---|---|---|---|---|---|---|---|---|---|
| | Pre | Post | Pre | Post | Time | Group | G × T | Time | Group | G × T | Time | Group | G × T |
| | 51.48 ± 3.85 | 55.03 ± 4.90 | 49.40 ± 3.75 | 48.19 ± 3.54 | 4.738 | 8.174 | 19.544 | 0.041 | 0.009 | <0.001 | 0.177 | 0.271 | 0.470 |

**Notes.**

Values are presented as mean ± standard deviation (SD).

Statistical analysis were performed using repeated-measures ANOVA for time, group, and group × time interactions.

$\eta^2$ represents partial eta squared, indicating effect size. Interpretation follows Cohen's conventions:

● Trivial (<0.2)    ● Small (0.2–0.5)    ● Moderate (0.5–0.8)    ● Large (>0.8).

$F$, test statistic; $P$, significance value; $\eta^2$, effect size.

**Table 5   Simple effect analysis of maximal oxygen uptake (VO$_2$max) in the lower-limb and control groups before and after training.**

| VO$_2$max/(ml/min/kg) | Time interaction | | | | Group interaction | | |
|---|---|---|---|---|---|---|---|
| | *F* | *P* | $\eta2$ | | *F* | *P* | $\eta2$ |
| Pre | 1.808 | 0.192 | 0.076 | Lower | 21.764 | <0.001 | 0.497 |
| Post | 15.371 | <0.001 | 0.411 | Contrast | 2.518 | 0.127 | 0.103 |

**Notes.**

This table reports the simple effect analysis of maximal oxygen uptake (VO$_2$max) before and after training in both groups.

Statistical analysis were performed using repeated-measures ANOVA for time, group, and group × time interactions.

$\eta^2$ represents partial eta squared, indicating effect size. Interpretation follows Cohen's conventions:

• Trivial (<0.2)   • Small (0.2–0.5)   • Moderate (0.5–0.8)   • Large (>0.8).

*F*, test statistic; *P*, significance value; $\eta^2$, effect size.

promoting metabolic adaptation and aerobic capacity. VO$_2$max, a core indicator of aerobic metabolic capacity, not only enhances the efficiency of aerobic energy supply but also aids in lactate clearance, accelerates phosphocreatine recovery, and delays fatigue. For endurance and speed-strength athletes, a high VO$_2$max contributes to optimizing the balance between energy systems, thereby improving overall athletic performance. The findings underscore the importance of simultaneously developing aerobic and anaerobic metabolic systems in training programs (*Archacki et al., 2024*). In terms of the energy metabolism system, this study found that after the HIIT intervention, the experimental group showed significant improvements in energy supply. Following the intervention, all key components of energy metabolism showed significant improvements. The phosphagen and glycolytic systems exhibited marked increases, reflecting enhanced anaerobic energy contribution during high-intensity efforts. Aerobic energy supply also improved notably, indicating a greater capacity for sustained oxidative metabolism. Overall, total energy supply demonstrated a substantial increase, highlighting the comprehensive enhancement of metabolic output induced by the HIIT protocol. These results indicate that HIIT effectively activates the phosphagen system to meet the energy demands of short-duration, high-intensity exercise while enhancing the adaptability of glycolytic and aerobic metabolism, thereby improving overall energy supply efficiency (*Sloth et al., 2013*). Consistent with the roles of different energy systems mentioned in the introduction, the phosphagen system plays a critical role in short-duration explosive actions such as starts, turns, and sprints. By contrast, the glycolytic and aerobic systems provide essential energy support during prolonged, sustained efforts in swimming (*García-Ramos et al., 2016*). HIIT, through the strategic alternation of high- and low-intensity phases, activates multiple energy systems and achieves efficient energy utilization. This mechanism is one of the key reasons for its significant effects observed in this study.

The lower-limb HIIT intervention led to significant enhancements across all energy systems. Specifically, the phosphagen, glycolytic, and aerobic pathways each exhibited notable increases in energy supply, demonstrating improved anaerobic and aerobic metabolic function. The overall rise in total energy supply was particularly prominent, aligning with previous research by *Eigendorf et al. (2021)* and further supporting the efficacy of HIIT in enhancing metabolic performance. This finding indicates that lower-limb HIIT effectively activated different energy supply pathways and enhanced the metabolic efficiency

of the athletes. In the experimental group, VO$_2$max showed a significant improvement following the HIIT intervention, reflecting enhanced aerobic capacity. In contrast, the control group did not exhibit any notable changes, highlighting the specific effectiveness of the training program.These findings demonstrate that HIIT has a significant effect on improving maximal aerobic capacity (*Song & Sheykhlouvand, 2024*). Significant interaction effects between time and group were observed across all energy supply pathways, including phosphagen, glycolytic, aerobic, and total energy metabolism. The effect was especially pronounced in the total energy supply, indicating that the training intervention had a substantial and differentiated impact on the experimental group compared to the control group. This finding indicates that the training effects became fully evident over time in the experimental group, confirming the effectiveness of lower-limb HIIT in optimizing energy metabolism and enhancing aerobic capacity (*Sousa et al., 2018*). The present study provides empirical support for the scientific training of swimmers (*Nugent et al., 2019*).

However, the results also revealed the interactions between different energy systems. While HIIT significantly increased the absolute energy supply across various metabolic pathways, the relative contribution percentages of each system showed no significant changes. This finding suggests that the primary effect of HIIT lies in enhancing the total energy supply rather than altering the relative contributions of individual energy systems (*Gastin, 2001*). This result is closely related to the contributions of different energy systems mentioned in the introduction: during short-duration, high-intensity exercise, the phosphagen system plays a dominant role, and the aerobic system contributes more significantly during prolonged, sustained activities (*Pyne & Sharp, 2014*). HIIT effectively harnesses the potential of different energy systems by repeatedly applying high-intensity stimuli within a short duration, combined with brief recovery periods, achieving efficient integration of energy metabolism (*Edge et al., 2005*). The design of this study highlights the application value of HIIT in lower-limb-specific training. In swimming, lower-limb power output and technical movements play a critical role in overall performance (*Zhou et al., 2024*). As mentioned in the introduction, lower-limb strength and metabolic adaptation are essential for supporting the start, turn, and acceleration phases in swimming (*Zamparo, Cortesi & Gatta, 2020*). Therefore, this study effectively enhanced energy supply and VO$_2$max through targeted lower-limb HIIT training, providing robust support for lower-limb performance in competitive swimming scenarios.

The present findings support and expand upon prior research demonstrating that high-intensity interval training (HIIT) enhances both aerobic and anaerobic energy systems in competitive athletes. For example, *Sperlich et al. (2010)* and *Dalamitros et al. (2016)* reported that HIIT led to significant improvements in VO$_2$max and metabolic capacity in swimmers, consistent with our findings. Notably, our results also revealed significant increases in glycolytic and phosphagen energy contributions, in line with findings from *Gosselin et al. (2012)* and *Almeida et al. (2021)*, suggesting that short-term HIIT can elicit robust adaptations in fast-twitch muscle fibers and anaerobic metabolism.

The physiological mechanisms behind these adaptations may involve increased mitochondrial density, enhanced oxidative enzyme activity, and improved muscle oxygen utilization. HIIT has also been shown to stimulate rapid glycogen resynthesis

and lactate clearance capacity, which may explain the improved glycolytic and phosphagen system performance observed in our study (*Edge et al., 2005*). In addition, neuromuscular efficiency, particularly in the lower limbs, may have contributed to the observed increase in energy system efficiency and VO$_2$max.

This study has several limitations. First, the sample size was relatively small and limited to elite swimmers from a single institution, which may affect the generalizability of the findings. Second, although we observed significant improvements in VO$_2$max and energy metabolism, we did not measure neuromuscular activity or lactate clearance directly, which could provide additional insights into the underlying mechanisms. Third, the menstrual cycle phases of female participants were not controlled, which may have influenced physiological responses. Future research should include larger and more diverse populations, incorporate neuromuscular and hormonal markers, and stratify data by sex and training status to better understand the full range of HIIT-induced adaptations.

In summary, this study demonstrated the significant effects of 4 weeks of lower-limb HIIT intervention in improving energy metabolism and VO$_2$max. HIIT not only significantly enhanced the energy supply efficiency of the phosphagen, glycolytic, and aerobic metabolic systems but also showed remarkable advantages in increasing VO$_2$max. These findings provide empirical support for the scientific and systematic training of competitive swimmers, further emphasizing the application value of HIIT in lower-limb-specific training. Coaches and researchers should fully leverage the benefits of HIIT, tailoring training programs to individual differences to maximize athletic performance.

## CONCLUSION

This study confirmed that a 4-week lower-limb high-intensity interval training (HIIT) program significantly enhanced energy metabolism and maximal oxygen uptake (VO$_2$max) in elite swimmers. Compared with traditional training, the HIIT protocol effectively activated the phosphagen, glycolytic, and aerobic energy systems, demonstrating a substantial improvement in total energy supply and aerobic capacity.

Beyond confirming the physiological effectiveness of lower-limb HIIT, the results emphasize its practical value in competitive swimming. Specifically, incorporating targeted HIIT sessions can improve metabolic efficiency without extending total training time, making it an efficient strategy for enhancing starts, turns, and sprinting power—all of which rely heavily on lower-limb output.

These findings provide evidence-based guidance for coaches to design individualized training interventions that maximize energy system adaptation. Future studies should explore long-term adaptations, neuromuscular mechanisms, and training optimization across different athlete populations and sexes.

### Funding

The authors received no funding for this work.

### Competing Interests

The authors declare there are no competing interests.

### Author Contributions

- Rangxi Jin conceived and designed the experiments, performed the experiments, analyzed the data, prepared figures and/or tables, and approved the final draft.
- Junhui Sun conceived and designed the experiments, performed the experiments, analyzed the data, prepared figures and/or tables, authored or reviewed drafts of the article, and approved the final draft.
- Na Jiang conceived and designed the experiments, authored or reviewed drafts of the article, and approved the final draft.
- Chao Chen conceived and designed the experiments, performed the experiments, prepared figures and/or tables, authored or reviewed drafts of the article, and approved the final draft.

### Human Ethics

The following information was supplied relating to ethical approvals (*i.e.*, approving body and any reference numbers):

This study was approved by the Ethics Committee of Shanghai University of Sport (102772021RT031).

### Data Availability

The raw measured values and the data on energy metabolism and maximal oxygen uptake tests are available in the Supplemental Files.

### Supplemental Information

Supplemental information for this article can be found online at http://dx.doi.org/10.7717/peerj.19788#supplemental-information.

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
