# Peer review of "Influence of 4-week lower extremity high-intensity interval training on energy metabolism and maximal oxygen uptake of elite swimmers"

_PeerJ, doi:10.7717/peerj.19788_

## Round 0.1 · original submission · Major Revisions

The reviewers have some major concerns regarding the manuscript. Reviewer 2 is particularly concerned about the reliability of the results. Please check this carefully and consider a further statistical analysis

Reviewer 1 ·

Basic reporting

The topic of the article is interesting. The Authors evaluated the effectiveness of a 4-week HIIT program on oxygen consumption and energy metabolism in élite swimmers. The article has been well designed and structured to answer the research questions. However, some suggestions could be discussed or clarified to improve the paper’s quality.

Introduction
Line 66: please delete a point between “…swimming performance” and the citation “(Sperlich et al., 2010a).
Lines 85-138: please synthesize these paragraphs.
Lines 139-149: please provide a better specification regarding the main hypothesis.

Experimental design

Methods
Line 164: please specify how the actual sample size was calculated, the randomization and allocation process, and the exclusion criteria.
Line 174: please clarify in deep the HIIT training program in terms of exercise selection, order and volume. The eight exercises the Authors proposed were conducted in-water or on dry-land? Lastly, why the intervention period lasted 4 weeks?
Line 188: is the lower-limb power cycle test a reliable and valid measurement for swimmers? Why did the authors evaluate energy metabolism and maximal oxygen uptake using a dry-land testing procedure rather than an in-water testing protocol?
Line 242: please some details about the repeated measures ANOVA analysis in terms of main effects, interaction effect size calculation.

Validity of the findings

Discussion
Line 376: please add a paragraph regarding future research directions and main limitations.

·

Basic reporting

In my personal opinion, I believe it needs major revisions to improve, before it can be published.
The abstract is clear and includes the objectives, design, methods, variables, main results, and most relevant conclusion.
Comment 1:
Line 36: “Statistical analyses were conducted using SPSS (version 25.0).” It would be better to write that “ANOVA repeated measures was carried out to examine interaction effects with data analysis and significance level was P < 0.05.” Please, correct it.
The introduction needs significant improvement to enhance clarity and flow, making it easier to read and helping the reader understand the topic.
General Comment:
In the first paragraph, consider introducing HIIT (High-Intensity Interval Training) by highlighting its application across various sports, including both team and individual, and emphasizing its effectiveness in enhancing aerobic fitness. In the second paragraph, discuss the body’s energy systems and how HIIT influences them. The third paragraph can focus specifically on swimming, explaining how HIIT contributes to improved performance in swimmers. Finally, in the fourth paragraph, clearly state the purpose of the research and outline any assumptions made during the design of the study. The introduction should be concise and well-structured, with a total length of approximately 700 to 850 words.
In addition, you should pay attention to some techniques in the way you write in the text as indicated below.
Comment 2:
Line 65 – 67: “Numerous studies have reported the effectiveness of high-intensity interval training (HIIT) in enhancing swimming performance (Sperlich et al., 2010a).” After you have mentioned numerous studies, you should put more than four references at the end of the sentence.
Comment 3:
Line 69 – 71: It is mentioned “Training within the near maximal range of 80% - 100% of maximal oxygen consumption (VO2max) or maximal heart rate (HRmax)”. However, the percentages of HRmax and VO2max are not equivalent. The percentage 80%–100% of HRmax corresponds to a lower percentage of VO2max. Unless you intended to refer to the same percentage range of HRmax and heart rate reserve (HRR). Please, clarify.

Experimental design

In Methods, methodology and techniques are adequate to reach the objectives of the study. While there are some details that are important.
Comment 4:
Line 167: “four females per group” Were these female participants' menstrual cycles recorded? We know from the literature that depending on the phase of the menstrual cycle, the follicular and luteal phases, women's performance is also affected. Of course, at this time the research was carried out and should be checked in its development in a future study.
Line 170 - 173: The body compositions of the participants are reported as the average but their standard deviation (±SD) is not reported. Please correct this.
The statistical analysis is described in detail and with great clarity but there are some questions such as:
Line 243: the program used was SPSS, version 25, Chicago USA, which is a statistical program and Microsoft Excel is an accounting program, which is not accurate in statistical analysis. Please clarify what was used for the reliability of the results.
Line 248: “P < 0.01 was considered highly significant”, It is not necessary to mention that; instead, reference should be made to the effect size, Cohen's η² index, and its interpretation, such as trivial (<0.2), small (>0.2 - 0.5), moderate (>0.5 - 0.8), large (>0.8).

Validity of the findings

The results are structured and are informative. The data should be recontroled
Comment 5:
In the tables, indicate the values, especially in η2 it should be evaluated. At the same time, check the values of η2 because in some parameters that I randomly calculated I found other values. Please do a detailed check of the results to be sure of the correct performance of the results.
Comment 6: Explain the notes at the bottom of every table and the legend format of figures at the bottom.
The discussion explains the results based on the literature and compare what was found in the present study with what exists in the literature.
Comment 7:
Τhe discussion could be more in-depth by comparing the results of this research with previous ones and providing an explanation through the mechanisms that cause these adaptations.
Comment 8:
Towards the end of the discussion you should mention some limitations.
Comment 9:
The conclusions are linked to proposals and no values are mentioned because they have been mentioned above.

Additional comments

I would like to congratulate the authors for the data they collected and this is an interesting study.
In the literature there are many references to HIIT implemented to individual and team sports. Some important reasons for mentioned the HIIT are the mitochondrial biogenesis, the improvement of aerobic fitness mainly in team sports and various ajustments. Some of the authors are Bangsbo, Bishop, Buchheit, Bogdanis, Callahan, Clemente, Dupont, Faude, Krustrup, Thomakos, e.t.c. You could find some of these to support your research. Υou must review and check the results carefully. This study is defined and meaningful.

---

## Round 0.2 · Minor Revisions

The manuscript is almost ready for acceptance once the few edits have been made

·

Basic reporting

I would like to extend my sincere congratulations to the authors for their dedication, ethical approach, and evident passion demonstrated throughout this research. Following the major revisions already undertaken, I believe that addressing a few remaining minor corrections will render the manuscript ready for publication.

General Comment:
Format the text and correct the font and various other details that need attention
The introduction is more clarity and flow.
The discussion is better structured but needs some corrections to flow into the text.

Experimental design

Comment 3:
Line 105: “effect size (f = 0.25)” f is written with a capital F. Please corrected it.
Line 174: The flow chart study should be a reference to the number of participants (N=24). It could be better structured and refer to the total number of participates, who were excluded from the study (due to injury or illness, etc.) and the division into experimental and control groups with N.
Line 215: I think that figure 3 is unnecessary because the Borg scale is known. If you want, please make a reference to it.

Validity of the findings

Comment 4:
Line 247: The values presented in Table 2 require verification, particularly the P-values, which should be reported accurately and in accordance with established statistical reporting guidelines. Report exact P-values (e.g., P = .032).
Comment 5:
Line 250: “Note. Values are presented as mean ± standard deviation (SD).” It should be written (±SD). Please, correct it.
Line 251: the term “Statistical comparisons” should be replaced with “Statistical analysis” to more accurately reflect the content of the section. Please, correct it.
Line 300: “analyses” should be replaced with “analysis”. Please, correct it.

Additional comments

Once again, I would like to express my respect for your positive comments and I hope that when you continue to contribute to science with new innovative research, I will feel proud to have collaborated with you.

---

## Round 0.3 · accepted · Accept

All comments and requirements have now been adequately answerd / fulfilled. Congratulations on the well-written and relevant article.

·

Basic reporting

no comment

Experimental design

no comment

Validity of the findings

no comment

Additional comments

no comment